# Segmentation-Free Outcome Prediction from Head and Neck Cancer PET/CT Images: Deep Learning-Based Feature Extraction from Multi-Angle Maximum Intensity Projections (MA-MIPs)

**DOI:** 10.3390/cancers16142538

**Published:** 2024-07-14

**Authors:** Amirhosein Toosi, Isaac Shiri, Habib Zaidi, Arman Rahmim

**Affiliations:** 1Department of Radiology, University of British Columbia, Vancouver, BC V5Z 1M9, Canada; arman.rahmim@ubc.ca; 2Department of Integrative Oncology, BC Cancer Research Institute, Vancouver, BC V5Z 1L3, Canada; 3Department of Cardiology, University Hospital Bern, CH-3010 Bern, Switzerland; isaac.shirilord@insel.ch; 4Division of Nuclear Medicine and Molecular Imaging, Geneva University Hospital, CH-1211 Geneva, Switzerland; habib.zaidi@unige.ch; 5Department of Physics & Astronomy, University of British Columbia, Vancouver, BC V6T 1Z1, Canada; 6Department of Biomedical Engineering, University of British Columbia, Vancouver, BC V6T 1Z3, Canada

**Keywords:** artificial intelligence, deep learning, outcome prediction, positron emission tomography, head and neck cancer

## Abstract

**Simple Summary:**

Head and neck cancer is a serious health concern that affects millions of people across the globe. Predicting how patients will respond to therapy is critical for providing optimal care. To make these predictions, one may first manually identify tumour boundaries on medical imaging in order to obtain the necessary information. Manually establishing tumour borders, however, is both costly and time-consuming, and prone to error and disagreement. Our AI-based research provides a unique method for predicting patient outcome that eliminates the need for manual delineation steps. Instead, we collect information from the patient’s whole head and neck area on PET scans as "looked upon" from a variety of angles. Our technique is faster, more consistent, and more precise than existing methods, with the potential to help doctors deliver better care to patients.

**Abstract:**

We introduce an innovative, simple, effective segmentation-free approach for survival analysis of head and neck cancer (HNC) patients from PET/CT images. By harnessing deep learning-based feature extraction techniques and multi-angle maximum intensity projections (MA-MIPs) applied to Fluorodeoxyglucose Positron Emission Tomography (FDG-PET) images, our proposed method eliminates the need for manual segmentations of regions-of-interest (ROIs) such as primary tumors and involved lymph nodes. Instead, a state-of-the-art object detection model is trained utilizing the CT images to perform automatic cropping of the head and neck anatomical area, instead of only the lesions or involved lymph nodes on the PET volumes. A pre-trained deep convolutional neural network backbone is then utilized to extract deep features from MA-MIPs obtained from 72 multi-angel axial rotations of the cropped PET volumes. These deep features extracted from multiple projection views of the PET volumes are then aggregated and fused, and employed to perform recurrence-free survival analysis on a cohort of 489 HNC patients. The proposed approach outperforms the best performing method on the target dataset for the task of recurrence-free survival analysis. By circumventing the manual delineation of the malignancies on the FDG PET-CT images, our approach eliminates the dependency on subjective interpretations and highly enhances the reproducibility of the proposed survival analysis method. The code for this work is publicly released.

## 1. Introduction

Head and neck cancer (HNC) is the seventh most common cancer worldwide. It encompasses malignancies involving the anatomic sites that compose the upper aerodigestive tract, including the oral cavity, pharynx, larynx, nasal cavity, paranasal sinuses, and salivary glands [1]. In 2016, over 1.1 million new cases and 4.1 million prevalent cases were reported, resulting in approximately 500,000 deaths worldwide [2]. Although the median diagnosis age for HNC is around 60 years, in recent years the incidence of these cancers has increased in adults younger than 45 years, mainly due to higher numbers of oropharyngeal cancers associated with oncogenic human papillomavirus (HPV) [3].

Accurate prognosis and staging is crucial for improving patient care, treatment planning, and disease management, ultimately leading to improved survival rates [4]. Recurrence-free survival (RFS) analysis is a common outcome prediction method used to assess the effectiveness of treatments for HNC. RFS measures the length of time after treatment during which the cancer does not recur. RFS plays a vital role in the management of HNC patients by providing information on the durability of treatment effects and helps to determine the optimal treatment strategy for individual patients [5]. RFS analysis is typically performed utilizing clinical and pathological features like tumor stage, grade, histological sub-type, and nodal involvement. However, relying solely on these factors may not fully capture the complex tumor heterogeneity and dynamic changes in the tumor micro-environment, which can limit the accuracy of prognostication [6].

Biomedical imaging methods such as computed tomography (CT), magnetic resonance imaging (MRI), and positron emission tomography (PET) can provide valuable information in various steps of HNC management [7]. Specifically, PET imaging plays an active role in staging, treatment planning, monitoring response, recurrence detection and surveillance for HNC patients [8,9]. Several quantitative parameters could be extracted from fluorodeoxyglucose (FDG) PET and CT images, providing information about the metabolic activity and aggressiveness of the tumor. This information can be helpful in predicting treatment response, disease progression, and patient survival. Some commonly used imaging biomarkers are Standardized Uptake Value (SUV) thresholds, such as SUVmean, SUVmax, SUVpeak, volumetric measures, such as Metabolic Tumor Volume (MTV) and Total Lesion Glycolysis (TLG) [10]. Additionally, the high-throughput extraction of radiomics features from PET-CT image modalities provides massive amount of hidden information that could help personalized management of the HNC [11,12].

Imaging biomarkers and radiomics features are tumor specific; i.e., they need to be extracted from the tumor region of intertest (ROI). Therefore, accurate manual segmentation of ROIs within medical images is a crucial step for extracting these features. Manual segmentation has been the fundamental step in various medical imaging analysis tasks [13,14]. However, it presents several challenges that can impact the accuracy, reproducibility, and efficiency of the analysis. Some of the key challenges of manual segmentation can be categorized as follows:**Manual segmentation is a subjective task.** Every expert can have their own observation of the imaging data. These differences in the observations lead to different interpretations of the same medical image, resulting in the inconsistencies in the delineation of regions of interest. The difference in the manual delineations made by different expert annotators is known as inter-observer variability. This phenomena can significantly affect the reliability and reproducibility of any analysis made based on the manual annotations, such as survival analysis [15].**Manual segmentation is highly time and labor intensive.** Unlike most of the publicly available non-medical image datasets, manual segmentation of biomedical image datasets needs significant human effort and expertise. The process of precisely delineating complex anatomical structures, tumors, or any other malignancies is tedious and prone to errors, requiring a great deal of time and precision [16] On the other hand, the availability of experts for performing manual segmentation is limited, especially in clinical settings with a large cohort of patients [17].**Manual segmentation is not easily scalable.** Being highly labor-intensive, as mentioned earlier, performing manual segmentation is not an easily scalable process to a large cohort of patients. This process is even more complex especially in case of cohorts composed of multiple centers [18]. Employing multiple annotators with different levels of expertise or training can amplify the inter-observer variabilities, limiting the generalizability of the analysis across different centers or different populations [19,20].**Manual segmentation of HNC tumors is prone to errors.** HNC tumors’ characteristics can be highly diverse. Despite being located in a relatively constrained anatomical location, HNC tumors can represent irregular shapes, high heterogeneity in intensity patterns, and highly variable contrast uptakes [21]. These can make the process of HNC tumor delineation more challenging, leading to higher uncertainties and errors in the process [22].**Manual segmentation of a single HNC tumor as a whole can be suboptimal.** Unlike tumors in other types of cancers, like prostate cancer primary tumors, where the expert annotator might delineate different regions of the tumor as a single ROI, HNC tumors are normally segmented as a whole as a single ROI. As mentioned earlier, this can neglect the distinct characteristics of different subregions within the tumor. However, different subregions of a tumor can contribute differently to the prognosis task [23,24].

To address these challenges, we propose a segmentation-free approach for recurrence-free survival analysis in HNC. By circumventing the manual segmentation step, our approach eliminates the dependency on subjective interpretations and enhances the reproducibility of the analysis. Instead, our method utilizes a deep learning-based object detection model to automatically crop the head and neck region in the FDG PET volumes, enabling a more efficient, objective, and reproducible analysis.

The main contributions of the present work can be summarized as follows:We proposed a completely segmentation-free approach for survival analysis in head and neck cancer patients using FDG-PET images.We utilized an object detection model to automatically crop the head and neck region in PET images, and as a result, eliminated the need for time and labour intensive task of manual tumor/metastatic lymph nodes segmentation.We introduced a novel technique for feature extraction from PET images, using multi-angle maximum intensity projection (MA-MIP) to enhance feature representation.We used pre-trained deep convolutional neural networks on large natural datasets, without any fine-tuning, to extract deep features from MA-MIPs.We explored of variety of feature fusion methods to aggregate information extracted from multiple MA-MIP projection views.We successfully demonstrated that our proposed segmentation-free survival analysis method outperforms existing approaches for recurrence-free survival analysis of the head and neck cancer patients on the HECKTOR 2022 challenge dataset.We comprehensively compared different CNN architectures, different pooling methods, and multiple fusion techniques to identify optimal combinations for the task of survival analysis in our proposed pipeline.We developed a highly reproducible and computationally efficient survival analysis pipeline that can potentially be deployed in settings with limited resources (GPUs, etc.).

In the subsequent sections, we describe the methodology employed to automatically crop the PET volumes (Section 2.2, Section 2.3 and Section 2.4), generate multi-angel maximum intensity projections of the cropped PET volumes (Section 2.5), extract deep features using a pre-trained deep convolutional neural network (Section 2.6, Section 2.7 and Section 2.8), and perform recurrence-free survival analysis (Section 2.9). We present the results of our study in Section 3, discuss their implications, and highlight the advantages of the segmentation-less approach in overcoming the challenges associated with manual segmentation in medical imaging analysis in Section 4.

## 2. Materials and Methods

### 2.1. Dataset

For this work, we used the training dataset of the third MICCAI Head and Neck Tumor segmentation and outcome prediction (HECKTOR2022) challenge [22,25]. The dataset comprises FDG PET/CT images, along with the clinical and survival data for a cohort of 489 patients with histologically proven H&N cancer located in the oropharynx region. Scans were gathered across 7 different medical centers, with variations in the scanner manufacturers and acquisition protocols [26] (See Table 1). All 489 patients underwent radiotherapy treatment planning and had complete responses to the treatment (i.e., disappearance of all signs of local, regional and distant lesions). CT images had an original median voxel size of 0.98×0.98×2.80 mm^3^, and the PET images had a median voxel size of 4.00×4.00×3.27 mm^3^. To ensure consistency, for all PET/CT images, both CT and PET volumes were resampled to a voxel size of 1.0×1.0×1.0 mm^3^ using a third-order spline method.

### 2.2. Method Overview

Here we describe our proposed segmentation-free outcome prediction method in detail (see Figure 1). First, we train the object detection model [27] on manually drawn bounding boxes containing the anatomical region of patients’ head and neck, on both coronal and sagittal maximum intensity projections of CT images, as described in Section 2.4. The trained network is then used for cropping the head and neck region of the corresponding registered FDG PET image volume of the patient, as described in Section 2.3 (See Figure 1A). Then, we employ a pre-trained CNN network with frozen weights to extract so-called “deep features” from 72 maximum intensity projections computed from 5-degree steps of axial rotations of the cropped head and neck PET volumes, resulting in 72 feature vectors per each patient, as elaborated in detail in Section 2.6 and Section 2.7. Next, we aggregate the extracted feature vectors of all views (MIPs) and process them through an ML pipeline for outcome prediction (Figure 1B). All the steps are described in detail in the following subsections. The code developed for this work is publicly available at: https://github.com/Amirhosein2c/SegFreeOP.

### 2.3. Automatic Cropping of the Head and Neck Region

HNC tumors mainly affect the upper aerodigestive tract. In the early-stage disease, the risk of distant metastasis is very low, where only approximately 10% of patients have distant metastasis at the time of diagnosis [28]. Given that, in order to focus on the main site of the disease affection, we employed a deep learning-based object detection model to automatically locate the anatomical upper aerodigestive region. To this end, we first generate a coronal and a sagittal maximum intensity projection of the CT scan of each patient for the whole dataset. Then, we manually annotate the anatomical head and neck region bounding boxes on both coronal and sagittal CT MIPs, using LabelImg annotating toolbox [29]. On the coronal MIP, we placed a bounding box vertically between the sternoclavicular (SC) joint (the link between the clavicle or collarbone, and the sternum, or breastbone) and the top of the head, or the top of the image, if the full skull is not covered in the CT scan image, and horizontally between the left and right acromioclavicular (AC) joint (the joint where the scapula or shoulder blade, articulates with the clavicle or collarbone). (See Figure 2). We target the clavicle bone to extend the oropharyngeal region down, mainly to contain all the primary tumors and the involved local lymph nodes in the anatomical region of interest. Since we defined the anatomical region of interest based on the regional bones, for generating coronal and sagittal CT MIPs, we clipped all CT voxel values in the corresponding range of hard tissue (bone) hounsfield units (700–2000).

### 2.4. Training the Object Detection Model on CT Projections

For the object detection architecture we used TridentNet [27]. TridentNet is a multi-stage deep learning-based 2D object detection model initially proposed for handling the highly variating object scales. Utilizing deformable convolution in its “Trident” blocks, TridentNet extracts multi-scale features with different receptive fields for different object scales from features of the same semantic layers. Based on our previous experience with this architecture for detecting very small and highly ambiguous objects in PET images [30,31], here we picked the same architecture for localizing the anatomical head and neck region in CT coronal and sagittal MIPs. To this end, first the training set was divided into 5 folds, 4 for training a network and 1 for testing. Both coronal and sagittal MIPs were fed to the networks for training, along with the manually made bounding boxes as ground truth. The predicted bounding boxes per each patient in 5 testing folds were then used for cropping the head and neck region on the corresponding PET volumes.

### 2.5. PET Multi-Angle Maximum Intensity Projections (MA-MIPs)

After cropping the head and neck region on the PET volume, the cropped regions underwent the feature extraction step. The common practice for extracting features from biomedical volumetric images are either extracting features directly from the the 3D volume, or the pixels of the transaxial layers. However, in our proposed method, we instead extract image features from multiple maximum intensity projections (MA-MIPs) of the axial rotations of the cropped PET volume. To this end, we rotate the cropped PET volume axially for 5 degrees step by step, and per each rotation a MIP is captured from the rotated volume. This step is repeated 72 times in every 5 degrees to cover one complete axial rotation (see Figure 1B). This enables us to leverage strong 2D convolutional neural network backbones pre-trained on large-scale public datasets like ImageNet, for extracting “deep” features.

### 2.6. Deep Feature Extraction from MA-MIP Projections

The vast size and diversity of ImageNet, with millions of natural images belonging to thousands of categories, have allowed training of models with robust image recognition capabilities that can effectively generalize across different visual domains, capturing complex patterns, textures, and structural relationships. By training a deep neural network on this dataset, the model learns to recognize features that are not specific to any particular class or modality, but rather generalize well across different visual domains.

Through transfer learning, the knowledge gained from one task can be transferred to another related task. By doing so, the model can leverage the knowledge it has acquired and adapt it to the new task, leading to improved performance compared to training a model from scratch. When using a pre-trained CNN backbone for biomedical image analysis, the early layers can capture low-level features like texture, color, and shape, while higher layers can capture more abstract features like organs, tissues, or lesions. This hierarchical representation enables the model to capture both local and global information in medical images, leading to better performance in various tasks [32].

In spite of the significant variability of medical images, mainly due to factors like patient demographics, acquisition protocols, and pathological conditions, the robustness of pre-trained CNN backbones on ImageNet can translate to medical images, helping the model to generalize across different populations, modalities, and diseases [33].

#### EfficientNet as the Feature Extraction CNN Backbone

Looking at the trends of the proposed convolutional neural network architectures on the ImageNet [34], from the emergence of AlexNet [35] in 2012 till the present, the proposed CNN architectures are becoming larger in order to gain higher accuracy. However, the more the architectures get larger, the higher the number of parameters they reach, so much that it seems they have hit the hardware limits [36].

The most common approach for scaling the convolutional neural networks is increasing the depth, in order to help the network to learn more complex features throughout the depth and to generalize well on new tasks. This comes with the cost of difficulties in terms of training such deep models due to the well-known problems such as gradient vanishing [37]. Despite the proposed techniques to overcome this issue, such as batch normalization and skip connections, the accuracy of deeper networks seems to saturate at a certain point [38].

Among the state-of-the-art architectures designed for ImageNet, EfficientNet [39] is an attempt to achieve higher accuracy by uniformly scaling the CNN architecture’s depth, width, and resolution, forming a compound scaling framework. In order to scale up the CNN network while keeping the number of parameters of the network low, as the main building block, EfficientNet uses Inverted Bottleneck Block previously proposed in MobileNetV2 architecture [40], along with squeeze and excitation optimization [41]. The essence of Inverted Bottleneck Block is using the Depthwise separable convolution layer instead of a regular convolution block, which splits the convolution operation into two separate operations, a depth wise convolution and a pointwise convolution. This splitting allows the model to perform the same convolution operation in each block with much less computation. First introduced in MobileNet architecture [42], the depthwise separable convolution enables the EfficientNet architecture to scale up and gain higher accuracy with a much lower number of parameters.

Building on top of a baseline model, called EfficientNetB0, seven scales of EfficientNet are provided with different uniform scales (EfficientNet B1–B7) which could achieve higher accuracies compared to the state-of-the-art CNN architectures with same depth on the ImageNet dataset, while having lower parameters and being faster. Besides the superior performance on the ImageNet, EfficientNets also achieved state of-the-art accuracy on many other widely used datasets by transfer learning, while having up to 21×less parameters compared to other CNNs [39].

The remarkable performance of the EfficientNet architecture in terms of transfer learning, along with the uniform scale-up of the baseline architecture, motivated us to apply this model for feature extraction in our work. To this end, we selected 3 different scales of the EfficientNet pre-trained on ImageNet, namely B1, B4, and B7 (small, medium, and large) for deep features extraction from the MA-MIPs of the cropped PET volumes, to compare the effect of model scale on the effectiveness of the extracted features for the outcome prediction task.

### 2.7. Deep Features Preparation

The output of the pre-trained EfficientNet used for feature extraction is the last convolutional block before the classifier head of the network which is in the shape of a 4D tensor (a batch of 3D tensors). In order to make a feature vector from the 3D feature tensor for each image in the batch, different methods are commonly used. The most common approach is to apply a global pooling layer on the output of the convolution block, either an average pooling or a max pooling, to summarize the feature tensor to a one-dimensional feature vector. The global max pooling operation essentially computes the maximum value over the entire height and width of a 2D convolutional block’s channel in the 3D tensor output’s depth. The global max pooling operation can be formulated as $\max_{h,w}\left(T_{c,h,w}\right)$.

Here, $T\in R^{C\times H\times W}$ represents the 3D tensor output of the 2D convolutional block, and the index *c* corresponds to the channels dimension. The indices *h* and *w* iterate over the spatial dimensions of the tensor. The “max” operation finds the maximum value among all the spatial locations in each channel for every instance. In global average pooling, similar to the global max pooling operation, the “mean” operation computes the average over all the spatial locations in each channel, formulated as $\frac{1}{h\times w}\sum_{i=1}^{h}\sum_{j=1}^{w}T_{c,i,j}$.

In this work we also extended the global pooling layer to the median and also standard deviation pooling in order examine different possible global pooling operators.

### 2.8. MA-MIPs Extracted Deep Features Fusion

Features extracted from MA-MIPs taken from different rotations of the cropped PET volume can be seen as the information extracted from different views of the PET volume. Therefore, combining the information extracted from different views could give a more informative set of information about the region.

In order to aggregate the feature vectors extracted from 72 MA-MIPs from different views of the cropped PET volume, feature fusion methods are applied. Multiple feature fusion methods are used in this work in order to aggregate feature vectores from multiple views into a single feature vector. The utilized methods can be categorized into three main groups: statistical summarizing methods, linear transformation methods, and non-linear transformation methods.

Statistical summarizing methods leverage simple statistical measures to aggregate information from the 72 feature vectors into a single feature vector. To this end, the index-wise (channel-wise) maximum, mean, median, and standard deviation of the feature vectors are computed after stacking all the 72 feature vectors of the different views. The maximum value represents the most prominent feature across all views, while the mean, median, and standard deviation capture different aspects of the overall statistical distribution of features from different views. These methods simplify the representation of the data by distilling it down to key statistical characteristics.

Linear transformation methods, on the other hand, encompass dimensionality reduction techniques applied index-wise (channel-wise) to the stacked feature vectors. Initially, the 72 feature vectors are stacked vertically to form a matrix, similar to the statistical summarizing methods, where each row represents a feature from one view, and there are 72 columns, each corresponding to a specific feature index (channel). Independent component analysis (ICA) is then individually applied to each column of the stacked feature vectors. This process summarizes the 72 cells within each feature index into a single value. By applying these linear transformations to each index separately, a reduced-dimensional representations for each feature index is obtained, in order to capture essential patterns and reduce dimensionality, while preserving unique characteristics specific to each view. This approach analyses how each feature contributes to the fused representation.

As for non-linear transformation methods, we used an auto-encoder. This learning-based method is applied on the concatenated 72 feature vectors to learn a compact, lower-dimensional representation in an unsupervised end-to-end manner. The auto-encoder comprises an encoder network that maps the input data into a reduced-dimensional latent space and a decoder network that reconstructs the input from this latent representation. By training the auto-encoder, it learns to capture the most salient features in the data while discarding noise and redundancies, resulting in an optimized fused feature vector that retains essential information. These three categories of fusion methods represent a well-rounded approach to fusing feature vectors from multiple views of the HNC PET images. Each category offers unique advantages and insights into the data, allowing for a comprehensive analysis that may enhance the predictive power of the proposed model for outcome prediction.

### 2.9. ML Prediction Pipeline for Recurrence-Free Survival Analysis

Fused multi-view features are then fed to an outcome prediction pipeline in order to perform recurrence-free survival prediction. Feature vectors of all 489 patients first underwent a standard scaling by removing the mean and scaling to unit variance. Samples were divided with a 80/20 ratio using nested 5-fold cross validation with 20 times repetition to evaluate the outcome prediction pipeline. Per each inner fold, highly correlated features were eliminated, then an independent component analysis (ICA) was applied on the remaining features to further reduce the dimensionality of the features and also to avoid over-fitting. The Cox proportional hazard survival method was employed on the reduced dimension features to predict the survival risk of each patient. Grid search was used for parameter tuning. The trained models were tested on each of the outer unseen testing folds and the performance of the trained outcome prediction models are reported in terms of the mean and the standard deviation of the concordance index (c-index) of the five unseen test folds. In the next section we report the results of our proposed pipeline for recurrence-free survival analysis, and evaluate the effect of using different methods for each step throughout the proposed segmentation-free outcome prediction pipeline.

## 3. Results

In this section we report the result of our proposed segmentation-free outcome prediction model for recurrence-free survival prediction of HNC patients using the multi-angle maximum-intensity projections of their FDG PET images. Table 2 summarizes the result of all possible combinations of the feature extraction backbones, pooling methods, and deep feature fusion methods, over all five test folds, in terms of mean and standard deviation c-index. Rows of Table 2 are divided into four sections each corresponding to one of four global pooling methods. On average, over all three different sizes of the feature extraction method (EfficientNet B1, B4, and B7), and using all six different feature fusion methods, global average pooling seems to be slightly more effective compared to other pooling methods, with the mean c-index of 0.659 compared to 0.653, 0.645, and 0.630 for standard deviation, median, and max pooling, respectively. However, the difference is minimal.

Comparing feature fusion methods, the independent component analysis (ICA) method seems to work poorly with respect to other fusion methods (0.519 on average over four pooling methods and three backbone sizes). Among two other fusion methods, on average, statistical summarization methods seems to outperform non-linear transformation method (auto-encoder) with the mean value over all five channel-wise methods equal to 0.674 compared to 0.668 for the auto-encoder. Between the four statistical summarization fusion methods, channel-wise maximum fusion with the mean c-index of 0.686 outperforms other methods with mean c-indices of 0.673, 0.672, and 0.668 for channel-wise standard deviation, mean, and median, respectively. Comparing the effect of using the EfficientNet as a feature extraction backbone with different sizes, looking at the average of the c-index values over all the proposed pooling and fusion methods, EfficientNet Large (B7) and EfficientNet Small (B1) with mean c-indices of 0.658 and 0.654 respectively, slightly outperform the EfficientNet Medium (B4) with the mean c-index of 0.630, with the large size performing minimally better than EfficientNet Small.

Picking the EfficienNet Large as the better performing feature extraction method, in order to compare the effectiveness of the proposed pooling methods, Figure 3 shows box plots of the c-index values of five test folds. Each plot shows the results using one of the four proposed pooling methods, for EfficientNet large, and all six different fusion techniques. Looking at the median and the variability of the box plots, the channel-wise maximum fusion method seems to perform slightly better than other statistical summarization methods and the auto-encoder method. However, for the global average pooling method, as the best pooling method among all proposed poolings, auto-encoder-based fusion seems to perform on par with the channel-wise maximum and mean fusion techniques. This is not true in other three diagrams showing results of methods using other pooling techniques. On the other hand, looking at the four statistical summarization methods and the non-linear transformation method (auto-encoder) in all four diagrams of the Figure 3, it seems that boxes are more compact with relatively higher median c-index values and lower variability compared to the boxes in other three diagrams corresponding to other pooling methods.

Likewise, Figure 4 shows in box plots the performance of the channel-wise maximum fusion technique and the auto-encoder as the two best fusion methods over all four different global pooling methods, over all five testing folds. Here as can be seen, global average pooling has a higher median c-index and lower variability compared to the other three pooling methods on both channel-wise maximum fusion and the auto-encoder.

Table 3 shows the results of the method proposed in [43] who won the best place in the HECKTOR 2022 challenge leaderboard for recurrence-free survival analysis task, on the training part of the dataset, in terms of mean, maximum, and minimum c-index over the test folds. Using global average pooling over the deep features extracted using EfficientNet with large size, all statistical summarization and non-linear transformation fusion methods outperform the result reported by [43] on the dataset.

We have also tried a few other well-known pre-trained CNN backbones for deep feature extraction in order to compare to the EfficientNet used in our proposed method. Table 4 summarizes the results of using nine other pre-trained CNN backbones in terms of mean c-index over the five test folds. For all these feature extractors, we used global average pooling and then the channel-wise maximum fusion method for the PFS prediction pipeline. While EfficientNet with small and large sizes seem to outperform all the other feature extractors, other backbones perform better or on par with EfficientNet medium. We report the result of all used CNN backbones, using all the proposed pooling methods and feature fusion methods in Table 5.

## 4. Discussion

Head and neck cancer (HNC) is a worldwide health concern that demands sophisticated diagnostic, prognostic, and management procedures. Recent advancements in AI have created new opportunities to enhance HNC management through processing different available imaging modalities [44]. AI-driven methods have been developed for tasks such as HNC cancer identification [45,46], tumor detection [47,48,49], classification of different types of HNC tumors and malignancies [50,51,52,53], automatic segmentation of HNC tumors [54,55,56], and outcome prediction [49,57,58] for HNC patients.

The present study proposed a segmentation-free outcome prediction method for HNC patients from PET/CT images. Instead of relying on manual segmentations, this work utilizes an object detection method that is trained to find the anatomical head and neck region of the patient on both coronal and sagittal MIPs of the patient’s CT scan. The benefit of using such a method for automatically cropping the anatomical head and neck region is that, compared to the time- and resource-demanding task of manual delineation of all the tumors and involved lymph nodes, detecting the anatomical site of the primary disease is a less time- and resource-demanding task. Defining the manual bounding boxes used in training the object detection method does not demand an expert physician. Constraining the entire space of the patient’s body to only the primary site of the disease has also the advantage of helping the feature extraction methods to focus more on the relevant information, in such a way that no delineation of the disease related uptakes on the FDG PET will be needed.

This point also can be seen in previous works on the second round of MICCAI’s HECKTOR challenge, where all the DICOM volumes were accompanied with 3D bounding boxes with size of 144×144×144 voxels containing the extended oropharyngeal region of the patients. The provided bounding boxes not only facilitated the use of 3D methods for automatic tumor segmentation due to the high computation and memory demands of such methods, but also made room for proposing outcome prediction methods extracting information directly from the given bounding box, instead of relying on the segmentation masks of the primary tumors, which resulted in the first and second best performing methods for the outcome prediction task [59,60] outperforming methods relying on predicting the segmentation masks of the primary tumors. It is worth noting that thanks to the recent success of methods such as TotalSegmentator [61], training an object detection network to crop the region is not necessarily needed. Instead, targeting the bones used in this work for defining the anatomical head and neck site (Section 2.3) one can use the segmentation masks of TotalSegmentator and use it to crop the ROI.

Beside using a bounding box of the entire disease site for extracting information (rather than manual delineations of the tumors/lymph nodes), in the present study, instead of extracting image features from the voxels of the 3D volumes of interest, or pixels of trans-axial slices of the 3D volume, feature extraction was performed on multiple maximum intensity projections of the VOI from different trans-axial angels. MIP projections from multiple axial angles can potentially enhance the representation of the malignancies (tumor and involved lymph nodes) and capture a broader view of its metabolic activity, while this information could be lost in common radiomics workflows. We have experienced the effect of enhanced representation of the malignancies in previous works for detection and segmentation of small metastatic lesions of biochemically recurrent prostate cancer [30,31]. The present work follows the idea of using multi-angle PET MIPs in cancer diagnosis tasks, but here specifically for the task of outcome prediction.

MIPs have been used in the literature for the task of outcome prediction. FDG PET coronal and sagittal MIPs were used in [62] for calculating two surrogate features to be replaced by their 3D equivalents, TMTV and Dmax, which were shown to be highly predictive in lymphoma cancer management. In [63], a CNN network was trained on MIP projections of FDG PET scans to predict the probability of 2-year time-to-progression (TTP). However both mentioned works used only the coronal and sagittal MIPs compared to 72 multi-angle MIPs in our work. In [62], MIPs were used indirectly as a means for estimating volumetric-based biomarkers through defining surrogate biomarkers, while our proposed method uses the information extracted from the MA-MIPs directly for the task of outcome prediction. Moreover, efforts in both [62,63] were either explicitly or implicitly aware of tumor segmentation masks, while our proposed method is completely segmentation-free.

Beyond the aim of our work in proposing a segmentation-free approach for HNC outcome prediction, there is a body of research on segmentation-free methods but for other aims. For instance, ref. [64] proposed a segmentation-free method for directly estimating the tumor volume and total amount of metabolic activity from patient PET images, which are two important imaging biomarkers in disease management and survival analysis of the patients. However, our present work bypasses the step of estimating the aforementioned biomarkers and directly predicts the outcome of the patients.

One of the main benefits of the proposed method in this work is the use of pre-trained CNN backbones with frozen weights for feature extraction. This property has multiple benefits. Firstly, using the pre-trained CNN backbones with frozen weights and without fine-tuning on the target dataset, makes the outcome prediction pipeline highly reproducible. This property is perfectly aligned with efforts such as the Image Biomarker Standardization Initiative (IBSI) for standardising the extraction of image biomarkers for high-throughput quantitative image analysis (radiomics) [65]. Moreover, it can also be beneficial in terms of computational resources.

Using the pre-trained CNN backbones as a pure transfer learning method, without the need for training on the target domain dataset, will make the outcome prediction pipeline highly portable, without need of high-end computational resources like GPUs. This helps the proposed methods be more generalizeable from limited training data and thus applicable in the clinic, using limited computational resources or embedded/edge devices. In any case, we plan to also study fine-tuning of the CNN backbones before feature extraction to evaluate the effect of domain adaptation on the target dataset on the task of outcome prediction.

Despite the promising results in our study, certain limitations exist that need to be addressed in future work: First, to better assess the generalizability of our proposed method, our experiments could benefit from an investigation of an external validation set. Secondly, given our inability to have access to the MICCAI HECKTOR challenge’s test set, we were unable to undertake a more extensive assessment against the baseline and all other competing approaches. This limits our ability to explicitly demonstrate the superiority of our suggested strategy over other solutions to the problem. Finally, while our deep learning-based feature extraction method has shown to be predictive, it still lacks explainability. Understanding the clinical or biological interpretation of these traits remains a challenge. Future research should be conducted on establishing tools for explainable AI and to confirm their importance in the context of head and neck cancer prognosis. Addressing these limitations in future investigations can help pave the way to clinical deployment of our proposed segmentation-free strategy.

## 5. Conclusions

We successfully developed an easy-to-perform segmentation-free outcome prediction method, specifically for predicting progression-free survival of head and neck cancer patients from their FDG PET-CT image volumes. We found that by using multiple rotation maximum-intensity projections of the PET images, and focusing the anatomical head and neck region guided by CT scan images of the patients, outcome prediction is not only feasible, but is also able to outperform conventional methods, without the need of manual segmentation of the primary tumors or the involved/metastatic lymph nodes by the expert nuclear medicine physicians.

Our proposed approach produced promising results; the top-performing configuration employed global average pooling, channel-wise maximum feature fusion method, and EfficientNet Large as the deep feature extractor, yielding a mean concordance index (c-index) of 0.719 ± 0.032 over five test folds. On the HECKTOR 2022 challenge dataset, our proposed approach performed better than the previous best performing method in the literature, with a mean c-index of 0.688. Additionally, the proposed segmentation-free method proved to be robust in a variety of configurations, employing several feature extractors, poolings, and feature fusion combinations that produced c-index values greater than 0.70.

The performance of our segmentation-free framework suggests higher reproducibility, by removing the subjectivity involved in the manual tumor delineation process, while simultaneously increasing the efficiency of the outcome prediction process. This approach has the potential to lead to better treatment planning and patient care by streamlining clinical procedures and providing more reliable prognostic information for patients with head and neck cancer.

## Figures and Tables

**Figure 1 cancers-16-02538-f001:**
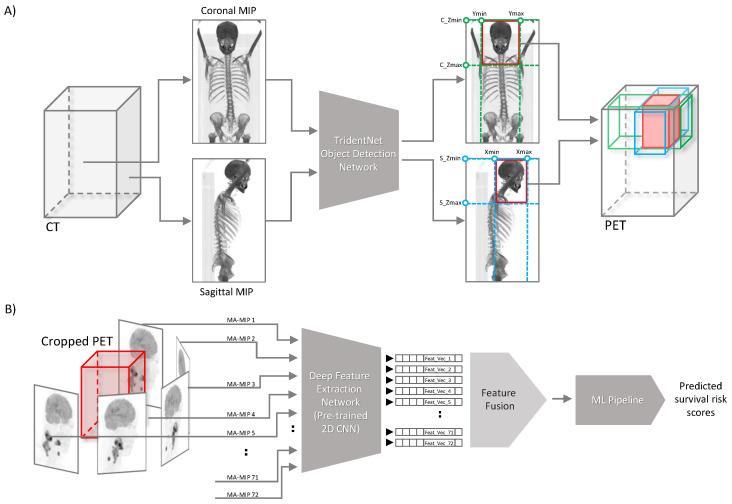
Overview of the proposed method for fully automated segmentation-free outcome prediction for head and neck cancer patients. (**A**) Overall representation of the automated head and neck region detection and cropping. (**B**) Extracting deep features from MA-MIP projections followed by feature fusion step.

**Figure 2 cancers-16-02538-f002:**
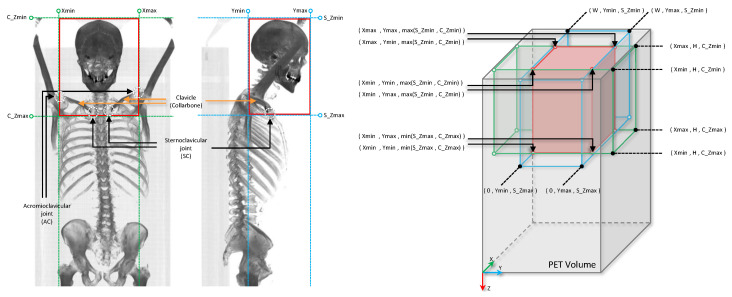
Visual summary of the proposed bounding box definition and cropping for locating the anatomical head and neck region on the PET volumes of the patients based on the coronal and sagittal MIP CT projections. Locating the bounding boxes on coronal and sagittal CT MIPs based on the AC and SC clavicle joints (**left**), and the 3D bounding box cropping on the PET volume based on the two 2D bounding boxes on the CT mips (**right**).

**Figure 3 cancers-16-02538-f003:**
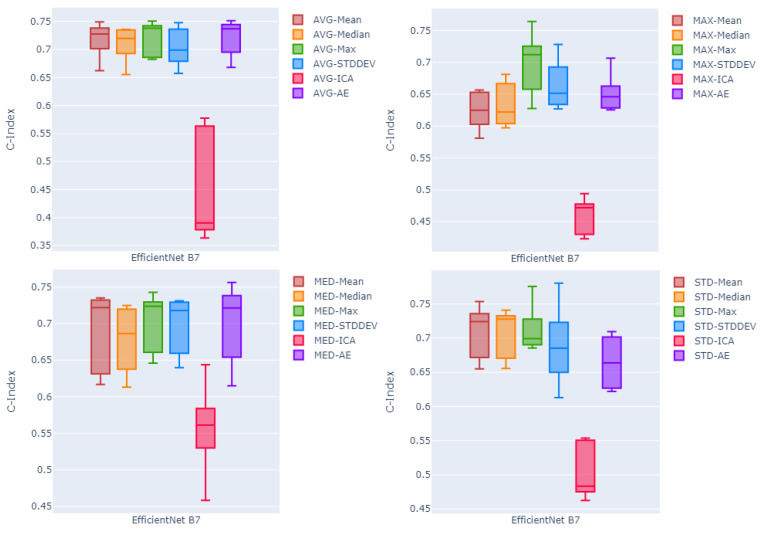
Box plots of the c-index values over all five test folds. Each plot shows the results using one of the four proposed pooling methods, for EfficientNet large as the feature extraction method, and all six different fusion techniques. Global average pooling (**top left**). Global max pooling (**top right**). Global median pooling, (**buttom left**), and global standard deviation pooling on the (**buttom right**).

**Figure 4 cancers-16-02538-f004:**
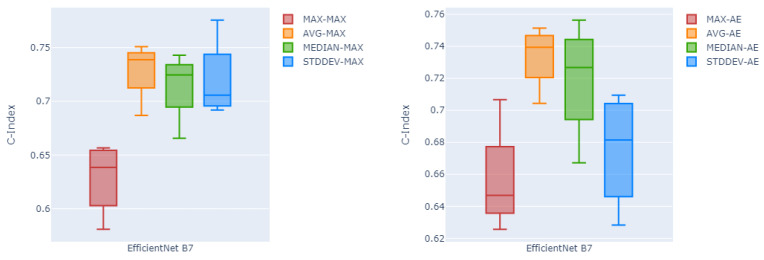
Box plots summarizing the performance of channel-wise maximum fusion technique (**left**) and the auto-encoder (**right**) as the two best fusion methods over all four different global pooling methods, and all five testing folds.

**Table 1 cancers-16-02538-t001:** Summary of the dataset used in this work.

Center	Acronym	PET/CT Scanner	Number of Cases
Hôpital général juif, Montréal, CA	HGJ	Discovery ST, GE Healthcare	55
Centre hospitalier universitaire de Sherbooke, Sherbrooke, CA	CHUS	GeminiGXL 16, Philips	72
Hôpital Maisonneuve-Rosemont, Montréal, CA	HMR	Discovery STE, GE Healthcare	18
Centre hospitalier de l’Université de Montréal, Montréal, CA	CHUM	Discovery STE, GE Healthcare	56
Centre Hospitalier Universitaire Vaudois, CH	CHUV	Discovery D690 TOF, GE Healthcare	47
Centre Hospitalier Universitaire de Poitiers, FR	CHUP	Biograph mCT 40 ToF, Siemens	44
MD Anderson Cancer Center, Houston, Texas, USA	MDA	Discovery HR, Discovery RX, Discovery ST, Discovery STE (GE Healthcare)	197

**Table 2 cancers-16-02538-t002:** This table summarizes the result of all combinations of feature extraction CNN backbones, pooling methods, and deep feature fusion methods, over all five test folds, in terms of mean and standard deviation c-index. Rows of the table are divided into four sections, each corresponding to one of four global pooling methods. Columns are also divided into three separate sections each corresponding to one fusion method, from left to right channel-wise statistical summarization methods, linear transformation method, and non-linear transformation method.

	Feature Fusion Methods					
**Feature Extractors**	**CW-Mean**	**CW-Median**	**CW-Max**	**CW-Std-Dev**	**ICA**	**AutoEncoder**
Global Average Pooling						
EfficientNet Small	0.706±0.05	0.707± 0.05	0.707±0.05	0.678±0.04	0.593±0.06	0.689±0.08
EfficientNet Medium	0.615±0.03	0.638±0.03	0.626±0.05	0.640±0.04	0.590±0.08	0.644±0.05
EfficientNet Large	0.718±0.03	0.710±0.03	0.719±0.03	0.705±0.04	0.455±0.10	0.721±0.03
Global Max Pooling						
EfficientNet Small	0.613±0.08	0.613±0.07	0.703±0.06	0.680±0.03	0.541±0.03	0.639±0.06
EfficientNet Medium	0.650±0.07	0.633±0.06	0.667±0.04	0.646±0.04	0.576±0.07	0.673±0.05
EfficientNet Large	0.625±0.03	0.634±0.03	0.697±0.05	0.665±0.04	0.459±0.03	0.651±0.03
Global Median Pooling						
EfficientNet Small	0.684±0.06	0.666±0.07	0.679±0.04	0.666±0.04	0.501±0.08	0.671±0.05
EfficientNet Medium	0.652±0.02	0.663±0.03	0.651±0.04	0.641±0.05	0.491±0.03	0.630±0.06
EfficientNet Large	0.688±0.06	0.678±0.05	0.701±0.04	0.697±0.04	0.556±0.07	0.698±0.06
Global STD-Dev Pooling						
EfficientNet Small	0.702±0.05	0.701±0.06	0.701±0.06	0.709±0.02	0.456±0.05	0.685±0.08
EfficientNet Medium	0.674±0.04	0.655±0.03	0.669±0.04	0.655±0.05	0.503±0.03	0.649±0.03
EfficientNet Large	0.708±0.04	0.706±0.04	0.713±0.04	0.689±0.06	0.506±0.04	0.665±0.04

**Table 3 cancers-16-02538-t003:** Comparison of the results of our proposed method, using global average pooling and EfficientNet large as feature extractor, to the baseline method [43] winner of the HECKTOR 2022 challenge for recurrence-free survival analysis task, on the training part of the dataset, in terms of mean, maximum, and minimum c-index over the test folds.

Outcome Prediction Method	Mean C-Index	Max	Min
L. Rebaud et al. [43]	0.688	0.732	0.642
EfficientNet Large + GAvgPool + CW-Mean (Ours)	0.718	0.749	0.662
EfficientNet Large + GAvgPool + CW-Median (Ours)	0.710	0.736	0.655
EfficientNet Large + GAvgPool + CW-Max (Ours)	0.719	0.751	0.682
EfficientNet Large + GAvgPool + CW-std-dev (Ours)	0.705	0.748	0.657
EfficientNet Large + GAvgPool + AutoEncoder (Ours)	0.721	0.742	0.668

**Table 4 cancers-16-02538-t004:** Result of using other architectures as feature extractors compared to EfficientNet in terms of mean c-index over the five test folds.

Feature Extractor	C-Index
InceptionResNetV2 + GAvgPool + CW-Max	0.641±0.06
InceptionV3 + GAvgPool + CW-Max	0.649±0.05
Resnet-152 + GAvgPool + CW-Max	0.662±0.06
VGG16 + GAvgPool + CW-Max	0.640±0.07
VGG19 + GAvgPool + CW-Max	0.635±0.03
Xception + GAvgPool + CW-Max	0.615±0.09
DenseNet-201 + GAvgPool + CW-Max	0.663±0.06
NASNet-Large + GAvgPool + CW-Max	0.644±0.04
ConvNeXt-Base + GAvgPool + CW-Max	0.660±0.06
EfficientNet Small (B1) + GAvgPool + CW-Max	0.707±0.05
EfficientNet Medium (B4) + GAvgPool + CW-Max	0.626±0.05
EfficientNet Large (B7) + GAvgPool + CW-Max	0.719±0.03

**Table 5 cancers-16-02538-t005:** Complete result of all used pre-trained CNN backbones, using all the four proposed pooling methods and five feature fusion methods from two groups of channel-wise statistical summarization fusions and non-linear transformation fusion method in terms of mean and std-dev of c-index over all five test folds.

Feat. Ext. Method	CW-Mean	CW-Median	CW-MAX	CW-STD-Dev	AutoEncoder
Global Max Pooling					
InceptionResNetV2	0.604±0.071	0.613±0.047	0.549±0.061	0.559±0.070	0.630±0.071
InceptionV3	0.640±0.056	0.627±0.041	0.643±0.049	0.638±0.037	0.637±0.050
Resnet	0.672±0.064	0.673±0.058	0.654±0.058	0.666±0.056	0.675±0.062
VGG16	0.644±0.051	0.644±0.044	0.686±0.032	0.631±0.066	0.599±0.026
VGG19	0.637±0.027	0.624±0.042	0.625±0.023	0.625±0.054	0.638±0.045
Xception	0.613±0.064	0.607±0.068	0.608±0.051	0.615±0.054	0.618±0.071
DenseNet201	0.685±0.060	0.679±0.053	0.689±0.054	0.612±0.037	0.683±0.026
EfficientNetB1	0.613±0.083	0.613±0.068	0.703±0.056	0.680±0.033	0.639±0.056
EfficientNetB4	0.650±0.068	0.633±0.055	0.667±0.036	0.646±0.044	0.673±0.053
EfficientNetB7	0.625±0.031	0.634±0.032	0.697±0.052	0.665±0.041	0.651±0.033
Global Average Pooling					
InceptionResNetV2	0.605±0.066	0.612±0.065	0.641±0.059	0.627±0.070	0.612±0.078
InceptionV3	0.635±0.052	0.630±0.049	0.649±0.048	0.654±0.043	0.651±0.050
Resnet	0.676±0.076	0.675±0.076	0.662±0.055	0.658±0.053	0.680±0.077
VGG16	0.641±0.052	0.624±0.056	0.640±0.072	0.614±0.092	0.655±0.061
VGG19	0.641±0.053	0.646±0.080	0.635±0.034	0.652±0.046	0.636±0.047
Xception	0.614±0.080	0.593±0.081	0.615±0.088	0.614±0.080	0.640±0.077
DenseNet201	0.668±0.061	0.665±0.066	0.663±0.064	0.653±0.055	0.683±0.074
EfficientNetB1	0.706±0.048	0.707±0.051	0.707±0.051	0.678±0.035	0.689±0.076
EfficientNetB4	0.615±0.028	0.638±0.029	0.626±0.051	0.640±0.041	0.644±0.048
EfficientNetB7	0.718±0.033	0.710±0.033	0.719±0.032	0.705±0.036	0.721±0.034
Global Median Pooling					
InceptionResNetV2	0.628±0.062	0.636±0.063	0.635±0.033	0.641±0.048	0.630±0.045
InceptionV3	0.632±0.038	0.655±0.044	0.660±0.036	0.645±0.065	0.628±0.057
Resnet	0.613±0.070	0.625±0.100	0.645±0.093	0.633±0.090	0.596±0.078
VGG16	0.571±0.038	0.574±0.025	0.557±0.041	0.556±0.049	0.582±0.055
VGG19	0.524±0.091	0.531±0.056	0.544±0.069	0.553±0.101	0.561±0.061
Xception	0.579±0.073	0.595±0.080	0.612±0.095	0.606±0.092	0.613±0.058
DenseNet201	0.644±0.066	0.653±0.068	0.634±0.088	0.618±0.065	0.665±0.061
EfficientNetB1	0.684±0.061	0.666±0.056	0.679±0.038	0.666±0.041	0.671±0.051
EfficientNetB4	0.652±0.017	0.663±0.027	0.651±0.040	0.641±0.045	0.630±0.055
EfficientNetB7	0.688±0.057	0.678±0.048	0.701±0.042	0.697±0.041	0.698±0.057
Global STD-Dev Pooling					
InceptionResNetV2	0.620±0.070	0.624±0.046	0.572±0.083	0.546±0.069	0.638±0.062
InceptionV3	0.634±0.634	0.635±0.058	0.643±0.040	0.638±0.040	0.639±0.066
Resnet	0.677±0.071	0.679±0.072	0.643±0.061	0.663±0.058	0.641±0.053
VGG16	0.654±0.046	0.648±0.035	0.661±0.064	0.630±0.051	0.628±0.042
VGG19	0.644±0.055	0.635±0.073	0.627±0.058	0.629±0.056	0.638±0.024
Xception	0.607±0.066	0.618±0.082	0.622±0.077	0.606±0.067	0.617±0.082
DenseNet201	0.670±0.049	0.664±0.032	0.657±0.045	0.655±0.031	0.682±0.045
EfficientNetB1	0.702±0.051	0.701±0.058	0.701±0.058	0.709±0.015	0.685±0.075
EfficientNetB4	0.674±0.041	0.655±0.032	0.669±0.044	0.655±0.047	0.649±0.026
EfficientNetB7	0.708±0.041	0.706±0.038	0.713±0.036	0.689±0.062	0.665±0.040

## Data Availability

The dataset was provided in the third edition of the head and neck tumor segmentation and outcome prediction challenge (HECKTOR 2022). Access to the dataset will be granted after approval of request. Please refer to the challenge webpage at this address: https://hecktor.grand-challenge.org/, accessed on 1 September 2023.

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
