# Peer review of "Segmentation-Free Outcome Prediction from Head and Neck Cancer PET/CT Images: Deep Learning-Based Feature Extraction from Multi-Angle Maximum Intensity Projections (MA-MIPs)"

_cancers, 2024, doi:10.3390/cancers16142538_

Round 1

Reviewer 1 Report

Comments and Suggestions for Authors

The author claimed to enhance the survival analysis method proposed by Ref 43 that predicts survival rate in head and neck cancers. In aiming to do so, they proposed a feature extraction technique that is based on multi-angle maximum intensity projection applied to PET and CT.  While the study is interesting, relevant and appropriate, there is no evidence that link their proposed method to the survival rate in head and neck patients. The paper is with clear flow, high quality figures/ tables and nice layout. However, the viewer has encountered some areas that needs to be improved before, including the aim, methodology and results. Enclosed below are some general and specific comments for the authors to consider.

General comments

1. You propose a free-segmentation method to crop the head and neck region in a PET image. How is this improving the prognosis and survival prediction?      

2. Re-write the title so it is more concise and includes CT as well, since the work was carried on both images PET/CT.

3. How does patient positioning affect the pixel values in both PET and CT. With head been extended or retracted impacting the tissue compression?  

4. Was the same data used for training used for testing i.e. 489?

Specific comments:

1.Abstrac Line 1: outcome prediction of what? Therapy, prognosis, survival etc. ….

2.Abstract Line 4: What are you referring to by volume? A lesion?

3. Abstract Line 7: you mean head and neck lesion instead of region?

4. Introduction Line 91: so the idea is to crop the entire head and neck region not to outline lesions in he head and neck region. This should be clearly state.  

5. Line 113: voxel size resampling increases the impact of partial volume effect. How was this addressed?

6. The claim is that the present work proposes a technique that is reliable, robots and accurate in predicting survival rate in head and neck cancers but the statistics in Table 2 are not strong, even if they show improvement to the prior study is clear.

7. Any limitations this study encountered?

Author Response

We wish to thank the reviewers for the valuable time and effort reviewing our work. Below we address points one-by-one, while also highlighting main changes in the main manuscript (additional editorial changes were also made for better clarity and readability though they are not highlighted in the main text). We believe and hope our replies and edits are now satisfactory, resulting in an improved manuscript. Thank you, and we remain open to any further suggestions. 

Comments and Suggestions for Authors 

The author claimed to enhance the survival analysis method proposed by Ref 43 that predicts survival rate in head and neck cancers. In aiming to do so, they proposed a feature extraction technique that is based on multi-angle maximum intensity projection applied to PET and CT. While the study is interesting, relevant and appropriate, there is no evidence that link their proposed method to the survival rate in head and neck patients. The paper is with clear flow, high quality figures/ tables and nice layout. However, the viewer has encountered some areas that needs to be improved before, including the aim, methodology and results. Enclosed below are some general and specific comments for the authors to consider. 

Response: We appreciate the valuable time, efforts and feedback on our work. We also appreciate the compliments that are work is interesting, relevant and appropriate, and has clear flow, high quality figures/tables and nice layout. We address points below one by one. In fact, our evaluation methods are directly related to survival prediction in head and neck patients as we utilized the C-index (concordance index) as a common metric for survival analysis used in numerous publications. We also elaborate further below. 

General comments 

  1. You propose a free-segmentation method to crop the head and neck region in a PET image. How is this improving the prognosis and survival prediction?

Response: Our proposed segmentation-free method can improve prognosis and survival prediction by eliminating the subjectivity and variability associated with the task of manual segmentation. By employing an object detection model able to perform cropping the head and neck region automatically, our method can ensure a consistent input for the machine learning-based survival analysis model across all patients. This approach captures the overall metabolic activity in the head and neck region of the patients from multiple views, including information from both primary tumors and involved lymph nodes, without relying on manual delineation of the aforementioned regions. This holistic multi-view approach may provide more comprehensive prognostic information than methods focusing solely on processing information extracted from manually segmented regions of interest. 

  1. Re-write the title so it is more concise and includes CT as well, since the work was carried on both images PET/CT.

Response: Thank you so much for the suggestion. We changed the title accordingly. The revised title now is “Segmentation-Free Outcome Prediction from Head and Neck Cancer PET/CT Images: Deep Learning-based Feature Extraction from Multi-Angle Maximum Intensity Projections (MA-MIPs)”. 

  1. How does patient positioning affect the pixel values in both PET and CT. With head been extended or retracted impacting the tissue compression?

Response: We would like to thank the reviewers for the interesting point raised. We acknowledge the fact that positioning of the patient’s head might affect the voxel values caused by the deformations of the soft tissue. From the anatomic point of view, however, this deformation will not affect the cropping of the head and neck region, given that enough samples are seen by the model during the training phase. Considering the targeted anchors for defining the head and neck region bounding boxes, density of the targeted bones won't change by any change in the position of the head and neck of the patient. On the other hand, since our method relies on extracted features from the multi-angle maximum intensity projections of the cropped PET volume, any compression or stretch in the tissues, specifically the high uptake lesions or involved nodes, won’t change the maximum value of the voxels belonging to these tissues. The only remaining concern could be regarding any change in the shape of the high uptake regions, which given the vast variety of the images taken from different centers, from different scanners, with the patients positioned in different poses, the trained survival analysis model should be generalizable enough to be invariant to the different positionings of the patient’s head. Below we depict coronal and sagittal MIPs of some sample patients in the dataset with different head positionings in support of this statement. 

  1. Was the same data used for training used for testing i.e. 489?

Response: For this work, we pursued a systematic approach in which we used a five-fold nested cross validation regimen applied to the same data (meaning of course that for any developed model, the training and testing portions were not mixed), and to avoid any accidental findings, the inner folds were repeated 20 times. This approach can help avoid over-fitting on the training set. To further clarify these points, we have added more explanations in the main manuscript in section 2.9. 

Specific comments: 

1.Abstrac Line 1: outcome prediction of what? Therapy, prognosis, survival etc. …. 

Response: Thank you so much for the comment. The defined outcome prediction task in this work, aligned with the task 2 of the MICCAI HECKTOR challenge, third edition, is in fact the task of Recurrence-Free Survival (RFS) prediction. The tool is generally applicable to different survival analyses, and later in the abstract, we clarify that in this particular work we are focusing on RFS. Overall, we have made edits to further clarify this point. 

2.Abstract Line 4: What are you referring to by volume? A lesion? 

Response: By volume, we refer to the 3D volumetric PET/CT images of the patients. We have changed “Volume” to “images” for clarification. 

  1. Abstract Line 7: you mean head and neck lesion instead of region?

Response: By region, here we mean the anatomical area of the head and neck. “region” was changed to “Anatomical area” accordingly, to help further clarify. 

  1. Introduction Line 91: so the idea is to crop the entire head and neck region not to outline lesions in he head and neck region. This should be clearly state.

Response: That is correct. We have now modified the Abstract, on line 7, to emphasize the point raised by respected reviewer. 

  1. Line 113: voxel size resampling increases the impact of partial volume effect. How was this addressed?

Response: We appreciate the reviewer's point on the partial volume effect. While resampling to a uniform voxel size is very common for consistent processing of the images, as pursued in this work, given data from different centers with multiple scanners each with different pixel sizes and slice thicknesses, we acknowledge it could amplify partial volume effect. However, as mentioned earlier, resampling voxel sizes to an isometric voxel, is a necessary step defined by the standard workflow of radiomics feature extraction (according to the Image Biomarker Standardization Initiative (IBSI)). In other words, regardless of the method used for feature extraction, voxel resampling needs to be performed when the data are acquired with different scanners. This is common practice, and the alternative, is a framework that entails significant other issues. In future work, an interesting area of research is to incorporate partial volume correction techniques or to explore optimal resampling methods that minimize this effect. 

  1. The claim is that the present work proposes a technique that is reliable, robots and accurate in predicting survival rate in head and neck cancers but the statistics in Table 2 are not strong, even if they show improvement to the prior study is clear.

Response: We appreciate the compliments for our technique. The comment pertaining to statistics is well-taken; we wish to add that the strength of the proposed segmentation-free outcome prediction method mainly relies on the fact that for performing outcome prediction, our method does not rely on any manual or automated segmentation of the tumors or metastatic lymph nodes. Given the multi-center data used in this work, acquired with different scanners, our method still shows robust and favorable results compared to the best performing method which is solely relying on multiple variations of segmentation methods. The improvements over that method we may not be able to demonstrate to be statistically significant, yet the technique is more efficient and less time- and labor-intensive. As also stated extensively in the introduction, being manual/automated segmentation-free, our method does not suffer from the inter-observer variability and other types of uncertainties associated with manual segmentations or inaccuracies associated with automatic segmentation methods. We have now extended the discussion section, and aimed to emphasize this point further. 

  1. Any limitations this study encountered?

Response: Thank you for this, which prompted us to improve the manuscript. Some key limitations of our study (which we now elaborate) include: 

  • Lack of external validation on an independent dataset. 
  • Not having access to the test set of the MICCAI HECKTOR challenge, to have a more comprehensive evaluation of our method with respect to the baseline. 
  • The need for further investigation into the interpretability of our extracted deep learning-based features.  

We have included a detailed discussion of these limitations in our revised manuscript in the discussion section. 

Reviewer 2 Report

Comments and Suggestions for Authors

The manuscript introduces an innovative, segmentation-free approach for outcome prediction in head and neck cancer (HNC) patients.

I have some comments:

Materials and Methods:

No specific details regarding patient consent. Your manuscript does not contain a complete IRB statement regarding ethics board approval. Original articles need to contain a statement about the Helsinki Declaration of 1975, as in the example given here: “This study was approved by the human subjects ethics board of XXXXX and was conducted in accordance with the Helsinki Declaration of 1975, as revised in 2013.

The methodology section lacks detailed descriptions of the segmentation process and the interface of the program used for automatic cropping of the head and neck region. Can the authors provide more information on these aspects?

Discussion:

The limitations and weaknesses of the study need to be added to provide a more comprehensive understanding of the study's reliability and reproducibility.

Author Response

We wish to thank the reviewers for the valuable time and effort reviewing our work. Below we address points one-by-one, while also highlighting main changes in the main manuscript (additional editorial changes were also made for better clarity and readability though they are not highlighted in the main text). We believe and hope our replies and edits are now satisfactory, resulting in an improved manuscript. Thank you, and we remain open to any further suggestions. 

Comments and Suggestions for Authors 

The manuscript introduces an innovative, segmentation-free approach for outcome prediction in head and neck cancer (HNC) patients. 

I have some comments: 

Response: We appreciate the valuable time, efforts and feedback on our work, and for the compliment that our work is innovative. Below we address comments one by one. 

Materials and Methods: 

No specific details regarding patient consent. Your manuscript does not contain a complete IRB statement regarding ethics board approval. Original articles need to contain a statement about the Helsinki Declaration of 1975, as in the example given here: “This study was approved by the human subjects ethics board of XXXXX and was conducted in accordance with the Helsinki Declaration of 1975, as revised in 2013. 

Response: As we also mentioned in the section “2.1 Dataset” in Methods and Materials, the dataset used in this work is the training dataset made publicly available (Upon Request, through filling a registration form of participation in the challenge) for the third MICCAI HECKTOR challenge of year 2022 (https://hecktor.grand-challenge.org/). As the registration form provided by the competition holders states: 

“Institutional Review Boards of all participating PROVIDER institutions permitted use of images and clinical data, either fully anonymized or coded, from all cases for research purposes, only. Retrospective analyses were performed in accordance with the relevant guidelines and regulations as approved by the respective institutional ethical committees with protocol numbers: MM-JGH-CR15-50 (HGJ, CHUS, HMR, CHUM) and CER-VD 2018-01513 (CHUV).  

For CHUP, the institutional review board approval was waived as all patients signed informed consent for use of their data for research purposes at diagnosis.  

For MDA, the ethics approval was obtained from the University of Texas MD Anderson Cancer Center Institutional Review Board with protocol number: RCR03-0800. 

For USZ, the ethics approval was related to the clinical trial NCT01435252 entitled “A Phase II Study In Patients With Advanced Head And Neck Cancer Of Standard Chemoradiation And Add-On Cetuximab”. 
For CHB, the fully anonymized data originates from patients who consent to the use of their data for research purposes.  

List of PROVIDERS: 

HGJ: Hôpital Général Juif, Montréal, CA 
CHUS: Centre Hospitalier Universitaire de Sherbrooke, Sherbrooke, CA 
HMR: Hôpital Maisonneuve-Rosemont, Montréal, CA 
CHUM: Centre Hospitalier de l’Université de Montréal, Montréal, CA 
CHUV: Centre Hospitalier Universitaire Vaudois, CH 
CHUP: Centre Hospitalier Universitaire de Poitiers, FR 
MDA: MD Anderson Cancer Center, Houston, Texas, USA 
USZ: UniversitätsSpital Zürich, CH 
CHB: Centre Henri Becquerel, Rouen, FR ” 

The methodology section lacks detailed descriptions of the segmentation process and the interface of the program used for automatic cropping of the head and neck region. Can the authors provide more information on these aspects? 

Response: In section “2.3. Automatic Cropping of the Head and Neck Region”, we have clarified the way we defined some anatomical landmarks that we used them for performing manual annotation of the anatomical head and neck region of the patients on both coronal and sagittal CT MIPs. We used Labeling tool (https://github.com/HumanSignal/labelImg), which is a well-known annotation tool in the object detection community. We performed manual annotation for all the patients in the dataset, then trained an object detection network on the manually annotated coronal and sagittal CT MIPs. Using a 5-fold cross validation step, we made sure the trained object detection model (TridentNet) is able to crop the anatomical head and neck region successfully. We have now added more details to the related section 2.3 in the paper, highlighted in blue, upon the reviewer's advice. Thank you. 

Reviewer 3 Report

Comments and Suggestions for Authors

1. Some parts of the paper look generated by AI/GPT such as lines 57~86, if not please attach images of AI content detector results in your cover letter which mentions lines written by humans.

2. All the main contributions of the papers should be included in the Introduction section.

Please make a Related Work section and review similar works (advantages and challenges) that were published last three years. Such as types of cancers (Neck cancers, brain cancers).

https://doi.org/10.3390/brainsci13091320

https://doi.org/10.3390/s22176501

https://doi.org/10.3390/cancers15164172

3. Dataset. Please make a Table to clearly describe your dataset based on the number of datasets, images and etc. Did you use data augmentation techniques to increase the number of images and overcome the overfitting problem?

4. Additionally, details are needed about how the data is divided into training, validation, and testing sets are lacking, such as whether the entire image is processed at once or if it's divided into patches.

5. A major issue lies in the presentation of results, which currently lacks sufficient detail. To assess clinical relevance, it's crucial for authors to visualize outcomes like detected tumors from the network and include statistical analysis to determine significance. Then medical discussion and medical expert evaluation must be added.

6. Additionally, authors must conduct an ablation study to evaluate the impact of transfer learning on the learning phase.

7. The conclusion part is very short and not included achieved outcomes efficiently.

Author Response

We wish to thank the reviewers for the valuable time and effort reviewing our work. Below we address points one-by-one, while also highlighting main changes in the main manuscript (additional editorial changes were also made for better clarity and readability though they are not highlighted in the main text). We believe and hope our replies and edits are now satisfactory, resulting in an improved manuscript. Thank you, and we remain open to any further suggestions. 

Comments and Suggestions for Authors 

Response: We appreciate the valuable time, efforts and feedback on our work by the reviewer, and below we address points raised one by one. 

  1. Some parts of the paper look generated by AI/GPT such as lines 57~86, if not please attach images of AI content detector results in your cover letter which mentions lines written by humans.

Response: We appreciate the reviewer for noticing and mentioning the concern regarding using AI for writing the lines mentioned above. We have used AI for summarizing the points we provided in the aforementioned lines in order to shorten the length of the introduction section. We have now revised the corresponding lines in the manuscript (highlighted in blue) and performed an AI content detection report. The report is attached to this note.

  1. All the main contributions of the papers should be included in the Introduction section.

Response: We now have a more elaborate contribution paragraph at the end of the Introduction section (Highlighted in blue). 

Please make a Related Work section and review similar works (advantages and challenges) that were published last three years. Such as types of cancers (Neck cancers, brain cancers). 

https://doi.org/10.3390/brainsci13091320 

https://doi.org/10.3390/s22176501 

https://doi.org/10.3390/cancers15164172 

Response: Thank you. We have now added a paragraph to the Discussion part according to the reviewer’s comment, elaborating the importance of such work, in a larger context, and included various added references including the ones mentioned above. 

  1. Dataset. Please make a Table to clearly describe your dataset based on the number of datasets, images and etc. Did you use data augmentation techniques to increase the number of images and overcome the overfitting problem?

Response: Thank you. We have now added a table (Table 1) detailing the dataset used for this work, the providing centers and the build of PET/CT scanners, as shown in Section 2.1  

  1. Additionally, details are needed about how the data is divided into training, validation, and testing sets are lacking, such as whether the entire image is processed at once or if it's divided into patches.

Response: We used the entire dataset using nested 5-fold cross validation, with 20 repetitions. Both inner and outer folds are split with an 80/20 ratio. The inner folds are repeated randomly for 20 times to avoid overfitting and any accidental findings. No augmentation or patching is used in feature extraction step, nor any over or under sapling is used in the ML survival analysis pipeline. To further clarify these points, we have now added more explanations in the main manuscript in the corresponding section (2.9). 

  1. A major issue lies in the presentation of results, which currently lacks sufficient detail. To assess clinical relevance, it's crucial for authors to visualize outcomes like detected tumors from the network and include statistical analysis to determine significance. Then medical discussion and medical expert evaluation must be added.

Response: Thank you. We have had several valuable discussions about readability and presentation of our work, and made a number of improvements. As for visualization, in the present work, the main gist of the technique is that it does not require detection or segmentation of tumors, and as such, no visuals are shown for that. However, in our figure 1, we elaborately show the visuals of the technique to perform anatomical delineation of the area of interest, on which segmentation-free outcome prediction is performed. Furthermore, we show how MA-MIPs, as a key innovative element of our work, are performed. As for statistical analysis, while we have performed comparison of our technique to other technique run on the same data, our lack of access to test data, as we now elaborate in a new ‘limitations’ discussion (end of discussion section), limits are ability to perform statistical comparisons. At the same time, we still find it very valuable to be able to perform comparison of our technique with a prior ‘winning’ technique, and to show our improved numbers on exact same data. Overall, we hope the new manuscript is satisfactory as we have addressed a number of comments by different reviewers.  

  1. Additionally, authors must conduct an ablation study to evaluate the impact of transfer learning on the learning phase.

Response: We have tried to further clarify many different points raised, and addressed a number of things. We feel that on this particular point, we can defer elaborate study of transfer learning to a future work. The present work outlines a new framework MA-MIPs, and we show very promising results only achieved by this technique. We set to also study different variations of this technique, as elaborated in different tables and figures, which in a sense also involve internal ablation study of how changes in the feature summarization method (Global Pooling methods) can affect the result of survival predictions (Please see Table 2). Additionally, we examined different feature fusion methods to see what is the effect of using different feature fusion methods in the accuracy of the predictions (Columns of Table 2). On the other hand, as also presented in Table 2, we have extracted features from EfficientNet with 3 different scales, to evaluate the effect of the scale of the deep feature extractor on the downstream task, which is in fact the task of recurrence-free survival analysis. And finally, in order to see how changing the architecture of the deep feature extraction method affect the down-stream ML pipeline for predicting the survivals, we replaced EfficientNet with 10 other deep feature extraction (pre-trained CNN) methods, with the associated results are presented briefly in Table 4 and more comprehensively in Table 5. Our upcoming/ongoing work also involves expanding this framework (MA-MIPs) to different applications, which we hope to publish. Thank you for your understanding. 

  1. The conclusion part is very short and not included achieved outcomes efficiently.

Response: We appreciate the comment regarding the conclusion part, and we have now added two more paragraphs to the conclusion part (highlighted in blue) and elaborated more on the achievements of the article.  

Round 2

Reviewer 1 Report

Comments and Suggestions for Authors

No new comments. The authors managed to satisfactory address most of comments.

Reviewer 2 Report

Comments and Suggestions for Authors

The author has well responded to all the reviewer's comments and incorporated the required changes as suggested by the reviewer. 

Reviewer 3 Report

Comments and Suggestions for Authors

The author has diligently addressed the reviewers' comments. There are no further comments on this paper.